# Selenoxides as Excellent Chalcogen Bond Donors: Effect of Metal Coordination

**DOI:** 10.3390/molecules27248837

**Published:** 2022-12-13

**Authors:** Sergi Burguera, Rosa M. Gomila, Antonio Bauzá, Antonio Frontera

**Affiliations:** Department of Chemistry, Universitat de les Illes Balears, Crta. De Valldemossa km 7.5, 07122 Palma de Mallorca, Spain

**Keywords:** selenoxides, metal coordination, chalcogen bond, X-ray structures, DFT calculations

## Abstract

The chalcogen bond has been recently defined by the IUPAC as the attractive noncovalent interaction between any element of group 16 acting as an electrophile and any atom (or group of atoms) acting as a nucleophile. Commonly used chalcogen bond donor molecules are divalent selenium and tellurium derivatives that exhibit two σ-holes. In fact, the presence of two σ-hole confers to the chalcogen bonding additional possibilities with respect to the halogen bond, the most abundant σ-hole interaction. In this manuscript, we demonstrate that selenoxides are good candidates to be used as σ-hole donor molecules. Such molecules have not been analyzed before as chalcogen bond donors, as far as our knowledge extends. The σ-hole opposite to the Se=O bond is adequate for establishing strong and directional ChBs, as demonstrated herein using the Cambridge structural database (CSD) and density functional theory (DFT) calculations. Moreover, the effect of the metal coordination of the selenoxide to transition metals on the strength of the ChB interaction has been analyzed theoretically. The existence of the ChBs has been further supported by the quantum theory of atoms in molecules (QTAIM) and the noncovalent interaction plot (NCIPlot).

## 1. Introduction

The interest by the scientific community in noncovalent interactions involving group 16 elements acting as electron acceptors is constantly growing [1,2,3]. Several experimental works have demonstrated the importance of chalcogen bonds (ChBs) in supramolecular chemistry [4], molecular recognition [5,6], crystal engineering [7], catalysis [8], and optoelectronics [9]. Moreover, several theoretical investigations have been reported with the purpose to comprehend the physical nature of ChBs [10,11,12]. Recently, the International Union of Pure and Applied Chemistry (IUPAC) has defined the chalcogen bond [13], following the principles used before for the definition of hydrogen and halogen bonds [14].

Selenoxides are mostly used in organic synthesis (selenoxide elimination) for the synthesis of alkenes and α,β-unsaturated carbonyl compounds from the corresponding saturated analogues [15]. They are also used in catalysis for the activation of hydrogen peroxide [16]. Moreover, selenoxides are also used as inhibitors of the δ-aminolevulinic acid dehydratase [17]. However, selenoxides have not been used in crystal engineering and supramolecular chemistry as ChB σ-hole donors.

The research work reported in this manuscript evidences the ability of selenoxides to form ChBs opposite to the Se=O bond (see Figure 1) by examining the Cambridge structural database (CSD). Moreover, it also demonstrates that the ChB is enhanced upon coordination of the selenoxide to a transition metal by means of DFT calculations. Four series of ChBs complexes have been optimized and compared using a variety of electron donors (Lewis bases and anions). Moreover, several computational tools were employed to study the ChBs, including molecular electrostatic potential (MEP) surfaces, to analyze the intensity of the σ-holes. Two methods based on the electron density topology were also used: the quantum theory of atoms-in-molecules (QTAIM) and the noncovalent interaction plot (NCIplot).

## 2. Results and Discussion

### 2.1. CSD Survey

The Cambridge structural database (CSD) [18] was initially inspected, since it is a large reservoir of chemical information. A total of 57 X-ray structures of selenoxide derivatives were found. Remarkably, in 50 out of 57 structures (87.7%) the selenium atom participates in at least one ChB that has a strong influence in the crystal packing. Although the number of structures where the selenoxide is complexed to a transition metal is reduced (5 out of 57), in all of them the selenium atom participates in a ChB, suggesting a strong ability of this type of compound to establish ChBs. It is also worth mentioning that 27 hits exhibited structure-directing ChBs (54%), where the σ-hole opposite to the Se=O is involved in the interaction, disclosing a slight preference for this σ-hole over the two Se–C σ-holes.

The chemical drawing of all structures studied herein are gathered in Figure 1. Figure 2 shows a selection of two X-ray structures exhibiting intermolecular ChBs that are relevant for the X-ray packing, which are LAWBIV [19] and MIHFAK [20]. In both structures, the existence of one dimensional supramolecular tapes can be observed, which are propagated through the formation of several ChBs. LAWBIV structure corresponds to dimethyl selenoxide exhibiting a trigonal-pyramidal structure. In the description of the solid state architecture by the original authors, only intermolecular C-H···O hydrogen bonds were mentioned to define the assembly shown in Figure 2a. These interactions, which are not shown in Figure 2a, are indeed present, complementing the ChBs indicated as dashed lines. The Se···O distances are 3.319 Å (opposite to Se=O) and 3.239 Å (opposite to Se–C), which are ~0.1 Å shorter than the sum of the van der Waals radii (ΣR_vdw_ = 3.45 Å) [21]. This fact along with the directionality of the O···Se=O interaction (~171°) strongly suggests the existence of directional σ-hole interactions. This is further confirmed by examining the crystal structure of bis(perfluoro-n-butyl)selenium(iv) oxide (MIHFAK) shown in Figure 2b, which lacks H atoms and consequently the possibility to form H-bonds. This compound also forms the Se···O contacts opposite to the Se=O bond and one of the Se–C bonds. The distance of the ChB opposite to the Se=O bond is longer than that in LAWBIV because, instead of the additional contribution of the CH···O contacts, an opposite effect (CF···O repulsion) occurs. In fact, the Se···O distance (3.454 Å) is almost identical to the ΣR_vdw_ (Se + O) value. In contrast, the ChB distance opposite to the Se–C bond is shorter (2.927 Å) due to the effect of the electron withdrawing F atoms that polarizes the Se–C bond and to the higher accessibility of this σ-hole (absence of O···F repulsion).

Figure 3 shows a selection of two X-ray structures exhibiting intramolecular ChBs that influence the conformation adopted by the selenoxide derivatives. In HEQTAZ [22] and HOPFAU [23], the ChB is established opposite to the Se=O bond, and in ROCJOJ [24], two intramolecular ChBs are established, one opposite to the Se=O bond and the other to the Se–C with identical distances. All of them are quite directional (≥170°) as is typical of σ-hole interactions. In all cases, the ChB distances are well below the sum of van der Waals radii, which are ΣR_vdw_(Se + Se) = 3.80 Å, ΣR_vdw_(Se + Cl) = 3.70 Å, and ΣR_vdw_(Se + N) = 3.50 Å. In HEQTAZ and HOPFAU, the short intramolecular distances are obviously influenced by the rigidity of the system (1,8-substituted naphthalene). In the case of HOPFAU, the original authors performed a theoretical study to analyze the Se···Se=O interaction in comparison to other selenyl-substituted 1,8-naphthalenes [23].

Figure 4 shows three X-ray structures exemplifying the structure-directing role of ChBs involving metal-coordinated selenoxides. The coordination of Se to Rh in BENYIF [25] and RUZCIY [26] and to Ru in YUTTOX [27] polarizes the Se=O bond and increases its ability to act as electron acceptor in σ-hole interactions. For instance, BENYIF forms self-assembled dimers in the solid state (Figure 4a), where two symmetrically equivalent Cl···Se=O interactions are established with a distance that is 0.3 Å shorter than ΣR_vdw_(Se + Cl) = 3.70Å. In these ChBs, the electron rich chlorido ligands interact with the Rh-coordinated diphenylselenoxide. In the case of RUZCIY, 2D supramolecular assemblies are formed in the solid state by means of multiple O···Se=O ChBs (see Appendix A for a representation of the 2D plane). A dimer extracted from this 2D assembly is shown in Figure 4b, exhibiting a quite short O···Se distance (3.075 Å) between the coordinated dimethylselenoxide moieties. The comparison of this distance to the one observed in the uncoordinated dimethylselenoxide shown in Figure 2a (3.319 Å) reveals a significant shortening of the ChB contact, thus suggesting a reinforcement of the ChB as a consequence of the metal coordination. Finally, it is worthy to highlight that in the X-ray structure of bis(6-(phenylselenyl)-6′-(phenylseleninyl-O)-(2,2′-bipyridine))-ruthenium(II) shown in Figure 4c (YUTTOX), the chloride counterions are located opposite to the coordinated Se=O bonds. An almost perfect linearity of the Cl···Se=O contact (176.6°) can be observed, along with the short distance (3.204 Å), which is 0.5 Å shorter than ΣR_vdw_(Se + Cl).

### 2.2. Theoretical DFT Study

A theoretical DFT study has been performed to study the ability of selenoxides to participate in ChBs and also the effect of the metal coordination, both in the σ-hole intensity and the interaction energies of some model compounds. Two different selenoxides were used (see Figure 2), dimethylselenoxide (**1**) and the perfluorinated analogue (**2**). Moreover, to analyze the effect of the metal coordination, AgCl was used for simplicity, due to the tendency of Ag(I) to form linear complexes. The effect of metal coordination has been studied in both the fluorinated and non-fluorinated selenoxides. As electron donors, some neutral Lewis bases (H_2_O, and sp, sp^2^, and sp^3^ hybridized N atoms) and anions were used in order to analyze the strength of the interaction depending also on the nature of the electron donor (see Figure 2 for the chemical drawings and numbering of the complexes).

#### 2.2.1. MEP Analysis

Initially, the MEP surfaces of compounds **1**–**4** were computed and represented (see Figure 5). It can be observed that the MEP maximum in dimethylselenoxide is located opposite to the Se=O bond (+30.3 kcal/mol). This σ-hole is also influenced (enhanced) by the nearby H atoms of the methyl group. The MEP value at the other two symmetric σ-holes is similar to the MEP maximum (+28.2 kcal/mol), thus revealing that all three σ-holes in compound **1** have similar ability to establish ChBs. The MEP minimum is located at the O atom (−48 kcal/mol), as expected. Upon coordination to AgCl (see Figure 5b, compound **3**), the MEP opposite to the Se=O significantly increases (+53.6 kcal/mol), indicating a strong polarization of the Se=O bond. The MEP at the symmetric σ-holes also increases (+38.0 kcal/mol). It should be emphasized that, upon coordination, the energetic difference between the σ-holes opposite to O and C atoms increases, thus increasing the preference of nucleophiles for the σ-hole opposite to Se=O.

An additional effect of the metal coordination is that the MEP value at the O atom becomes less negative (−22.6 kcal/mol), thus decreasing its nucleophilicity and ability to act as an electron donor. The MEP of compound **2** (see Figure 5c) shows different types of σ-holes (a total of five σ-holes), three opposite to the covalent bonds of the Se atom and two additional at the C atoms opposite to the C–F bonds. The deepest one is opposite to the Se=O bond (+36.4 kcal/mol), followed by those opposite to the C–F bonds (+34.5 kcal/mol), and finally those opposite to the Se–C bonds (+29.5 kcal/mol). The MEP at the O atom is −30.7 kcal/mol, less negative than that in compound **1**, as expected by the introduction of the fluorine atoms. The comparison of the MEP surfaces of compounds **1** and **2** discloses that the electrophilicity of the Se atom increases moderately (MEP value increases from 30.3 kcal/mol in **1** to 36.4 kcal/mol in **2**) whilst the nucleophilicity of the O atoms decreases significantly (MEP value changes from −48.0 kcal/mol in **1** to −30.7 kcal/mol in **2**). Therefore, the fluorination increases the ability to establish ChBs but globally decreases the ability to form homodimers (Se=O···Se=O) due to the larger reduction of the nucleophilicity of O than the increment of electrophilicity of Se. This agrees well with the geometric features of the X-ray structures of LAWBIV and MIHFAK where the Se=O···Se=O distance is shorter in the non-fluorinated structure. The MEP surface of compound **4** (Figure 5d) evidences that the electrophilicity of the σ-holes increases significantly upon coordination with respect to compound **2**. Moreover, the MEP at the O atom becomes very small (−6.2 kcal/mol) due to the double effect of the fluorination and the coordination to the Ag atom.

#### 2.2.2. DFT Energetic and Geometric Studies

Table 1 gathers the interaction energies, equilibrium distances, and Y:···Se=O angles of the ChB complexes **5**–**28**. The inspection of the energetic results of Table 1 discloses that the fluorinated ChB donor **2** (complexes **11**–**16**) exhibit slightly stronger interaction energies with respect to the non-fluorinated dimethylselenoxide **1** (complexes **5**–**10**), in line with the small differences between the σ-holes of compounds **1** and **2** (see Figure 5). The interaction energies involving neutral Lewis bases range from −4.2 kcal/mol to −7.0 kcal/mol for the complexes of **1** and **2**, and those with anionic donors are significantly more favorable (−20.2 to −30.1 kcal/mol) due to the stronger electrostatic attraction. Interestingly, a difference between the non-fluorinated and fluorinated complexes was found in the Y:···Se=O angle (α) that is closer to linearity in the fluorinated ones (171.4 to 178.4°). A significant enhancement of the interaction energies is observed upon coordination of AgCl in both series. As a representative example, the water complex changes from −4.2 kcal/mol in **1** + H_2_O (complex **5**) to −6.5 kcal/mol in **3** + H_2_O (complex **17**) and from −5.1 kcal/mol in **2** + H_2_O (complex **11**) to −7.2 kcal/mol in **4** + H_2_O (complex **23**). Moreover, a concomitant shortening of the ChB distances is also observed. It is worthy to emphasize that the distances in complexes **9** and **10** (involving the non-fluorinated **1**) are longer than the ΣR_vdw_. Moreover, both complexes present the interaction angles (Y:···Se=O) of the three series of complexes that deviate more from linearity (~163°, see Table 1). This suggests that in these complexes, H-bonding interactions with the methyl groups dominate with respect to the chalcogen bond, as further discussed below.

For the NH_3_ complexes, we have also computed the geometries and energies using the post-Hartree–Fock method Møller–Plesset (MP2) to validate the DFT method used herein. These values are given in parenthesis in Table 1. It can be observed that the geometric features of the complexes are almost identical, thus giving reliability to the DFT geometries and confirming the tendency of selenoxides to establish ChBs. The MP2 energies (see Table 1) are slightly less negative than the DFT ones (between 1 and 3 kcal/mol) suggesting that the DFT method slightly overestimates the interaction energies. However, the tendency is the same, thus validating the conclusions derived from the DFT analysis.

#### 2.2.3. QTAIM and NCIPlot Study

The chalcogen bonding complexes were further characterized using two computational tools based on the topological analysis (QTAIM and NCIPlot). The NCIplot is based on the electron density (ρ) and is adequate for revealing NCIs in real space via the representation of the reduced density gradient (RDG) isosurface. Moreover, the (signλ_2_)*ρ value is mapped onto the surface (λ_2_ is the middle eigenvalue of the Hessian) using a color scale. In this manuscript, green and blue colors are used for weak and strong interactions, respectively. Figure 6 shows the QTAIM/NCIplot representation of the complexes of compound **1** (dimethylselenoxide) and those of compound **3** (dimethylselenoxide coordinated to AgCl) in Figure 7. The representations for the rest of complexes are given in Appendix A (ESI). For the complexes of **1**, it is interesting to highlight that for all neutral complexes apart from **6** (acetonitrile), the electron donor atom is connected to the selenoxide via three bond critical points (CP, red spheres) and bond paths connecting the O,N atom to the Se and two H atoms of **1**. In the case of complex **6**, a CP and bond path connect the N atom to the Se. However, two green RDG isosurfaces are located between the N atom and the H atoms, revealing the existence of some attractive interaction (CH···N). This combination of interactions likely explains the small difference between the fluorinated (only ChBs, see Appendix A) and non-fluorinated (ChB + HB) complexes. In case of the anionic complexes **9** and **10**, the anion is connected to two H atoms via two bond CPs and bond paths. Interestingly, there is not any bond CP connecting the Se atom to the anion, in line with the long Se···Cl,Br distances and worse directionality compared to the neutral complexes. In these complexes, the ChB is only revealed by a green RDG isosurface located between the anion and the Se (see Figure 6e,f). This strongly suggests that both complexes **9** and **10** are dominated by H-bonds.

The QTAIM/NCIplot analyses of Figure 7 (complexes of **3**) show a markedly different behavior compared to the complexes of compound **1**. Only complexes **17** and **18** (water and acetonitrile as donors) exhibit three bond CPs and bond paths connecting the O,N atom to the Se and two H atoms. For the rest of complexes, the electron rich atom is only connected to the Se atom by a bond CP and bond path. The NCIplot shows, for the complexes **20**–**22**, green RDG isosurfaces between the electron rich atom and the methyl H atoms that suggest the existence of weak CH···N,Cl,Br contacts. Therefore, the selenoxide coordination to a metal center favors the ChB over the HBs, likely due to the larger polarization of the Se=O bond compared to the C–H bonds upon coordination. In compounds **21** and **22**, the presence of dark blue isosurfaces between the Se and the halide further corroborates the large enhancement of the ChB.

#### 2.2.4. Natural Bond Orbital (NBO) Analysis

A common feature of σ-hole interactions in general and chalcogen bonds in particular is the existence of an LP(Y:)→σ*(Ch–X) donor-acceptor interaction. This characteristic orbital interaction is appropriate to differentiate σ-hole bonding from other donor-acceptor interactions, such as coordination. A convenient methodology to study donor-acceptor orbital interactions is the natural bond orbital (NBO) through the second-order perturbation analysis (see Section 3 below). This methodology enables the identification of the donor-acceptor orbitals and the associated stabilization energy. In this section, we have performed the analysis for the ammonia complexes **20** and **26**, to analyze the effect of the metal coordination. The NBOs are represented in Figure 8 accompanied by the stabilization energies (E^(2)^ values). As expected, the analysis shows the typical LP(N)→σ*(Se–O) orbital interaction with concomitant stabilization energies of E^(2)^ = −1.3 kcal/mol and E^(2)^ = −2.7 kcal/mol for complexes **20** and **26**, respectively. It can be clearly observed that the filled LP orbital of the N atom points perfectly to the antibonding σ*(Se–O) orbital, thus confirming the σ-hole nature of the interaction. The orbital contribution is significant in both compounds compared to the total interaction energy, revealing that charge transfer effects are relevant. In addition, the larger stabilization energy for the complex where the selenoxide is coordinated to the Ag atom indicates that the enhancement of the interaction energy is not only due to electrostatic effects (more intense σ-hole) but also to an enhancement of the charge transfer.

## 3. Materials and Methods

The optimization of the geometry of compounds and complexes studied herein were done using the Turbomole 7.2 program [28]. For the calculations, the PBE0 functional [29] was used in combination with Grimme’s D3 [30] dispersion correction and the Weigend’s basis set def2-TZVP [31,32]. This level of theory was used before to study σ-hole interactions [33,34,35]. The MEP surfaces were plotted using the wavefunctions generated at the PBE0-D3/def2-TZVP level of theory and at the 0.001 a.u. isosurface. The topological analysis of ρ(r) (electron density) was carried using the quantum theory of atoms in molecules (QTAIM) [36] and supplemented with the noncovalent interaction plot index (NCIplot) [37] by representing the reduced density gradient (RDG) isosurfaces. The RDG isosurfaces were represented using the VMD program [38]. The s = 0.5 a.u.; cut-off ρ = 0.04 a.u.; color scale −0.03 a.u. ≤ sign(λ_2_)ρ ≤ 0.03 a.u. settings were used for the RDG plot. The natural bond orbital (NBO) [39] study was carried out using the NBO7.0 program [40] at the same level of theory.

## 4. Conclusions

This manuscript provides evidence that ChBs involving selenoxides are important in the solid state and, consequently, they can be rationally used in crystal engineering and supramolecular chemistry. The DFT study discloses that the energies in non-coordinated selenoxides are weak (similar to H-bonds for the neutral donors) and that the interaction is significantly enhanced upon coordination of the selenoxide to a transition metal. The combined QTAIM/NCIplot analysis shows that in the non-fluorinated complexes, H-bonds also participate in the interaction, which is dominant in the anionic donors. Upon coordination to AgCl, the ChB becomes dominant in all complexes. The MEP analysis discloses the presence of different types of σ-holes opposite to the Se–C, Se=O, and C–F in some cases. Finally, the participation of the antibonding σ*(Se=O) orbital is evidenced by the NBO analysis, thus strongly supporting the σ-hole nature of the interaction.

## Data Availability

Not applicable.

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
