# Peer review of "Selenoxides as Excellent Chalcogen Bond Donors: Effect of Metal Coordination"

_molecules, 2022, doi:10.3390/molecules27248837_

Round 1
Reviewer 1 Report
The manuscript reports an interesting theoretical study on the ability of selenoxides to act as chalcogen bond donor compounds. In this paper, the authors present the investigation of four species of selenoxide compounds, which act as electron acceptors in the chalcogen bons, with several different electron donor species. The selenoxide compounds include dimethylselenoxide, its perfluorinated analogue, as well as both of the previous compounds coordinated to AgCl. The manuscript is very well written, well organized and the main conclusions are well supported by the theoretical results presented in the study. In particular, the theoretical evidences demonstrated can be very relevant to the solid-state fundamental and application research, as the authors refer, which I think is definitely one of the most important strengths of this work.
However, before the manuscript can be accepted to publication in Molecules, in my opinion, there are several minor corrections that must be performed:
Some minor English correction can be done (e.g.: line 25 “has been constantly growing” instead of “is constantly growing”; line 53 “since it is a large” instead of “since it is large”).
I believe that, in line 61, the sentence must be changed to “disclosing a slight preference”, since the σ-hole opposite to the Se=O interactions are indeed a majority of the 57 studied structures, however by a short margin.
Additional, in my opinion, a short but better description is need to present the several compounds whose structure was studied in this work. Hence, the first time that molecular structures such as LAWBIV, MIHFAK and HEQTAZ (only to name a few) appear in the text, a brief introduction about the compounds that these structures represent must be included.
In line 150, the term “more positive” should be changed to “less negative”. This also happens several other times throughout the manuscript (e.g. line 156).
In lines 169 and 266, “silver” must be corrected to “silver atom”.
In Table 1, next to the equilibrium distances column, I think it would be interesting to include an additional column with the van der Waals radii of the corresponding interaction, for a better discussion of the presented results.
Finally, the formatting of the Figures must be redone, as they are not in line with the text.
To conclude, it is my opinion that the manuscript can be accepted for publication in Molecules after a minor revision is performed.
Author Response
First of all, we would like to thank this reviewer for his/her careful reading of the manuscript, corrections and suggestions. The changes made and our replies follow.
The manuscript reports an interesting theoretical study on the ability of selenoxides to act as chalcogen bond donor compounds. In this paper, the authors present the investigation of four species of selenoxide compounds, which act as electron acceptors in the chalcogen bons, with several different electron donor species. The selenoxide compounds include dimethylselenoxide, its perfluorinated analogue, as well as both of the previous compounds coordinated to AgCl. The manuscript is very well written, well organized and the main conclusions are well supported by the theoretical results presented in the study. In particular, the theoretical evidences demonstrated can be very relevant to the solid-state fundamental and application research, as the authors refer, which I think is definitely one of the most important strengths of this work.
However, before the manuscript can be accepted to publication in Molecules, in my opinion, there are several minor corrections that must be performed:
Some minor English correction can be done (e.g.: line 25 “has been constantly growing” instead of “is constantly growing”; line 53 “since it is a large” instead of “since it is large”).
Reply: Fixed, thanks!
I believe that, in line 61, the sentence must be changed to “disclosing a slight preference”, since the σ-hole opposite to the Se=O interactions are indeed a majority of the 57 studied structures, however by a short margin.
Reply: Done, thanks!
Additional, in my opinion, a short but better description is need to present the several compounds whose structure was studied in this work. Hence, the first time that molecular structures such as LAWBIV, MIHFAK and HEQTAZ (only to name a few) appear in the text, a brief introduction about the compounds that these structures represent must be included.
Reply: Thank you for this comment (also a suggestion of referee 2). We have added a new Scheme in the manuscript to facilitate the reader following the discussion.
In line 150, the term “more positive” should be changed to “less negative”. This also happens several other times throughout the manuscript (e.g. line 156).
Reply: Fixed, thanks!
In lines 169 and 266, “silver” must be corrected to “silver atom”.
Reply: Fixed, thanks!
In Table 1, next to the equilibrium distances column, I think it would be interesting to include an additional column with the van der Waals radii of the corresponding interaction, for a better discussion of the presented results.
Reply: Thank you for this suggestion. Done
Finally, the formatting of the Figures must be redone, as they are not in line with the text.
Reply: This is the format imposed by the journal.
Reviewer 2 Report
This paper which theoretically studies the chalcogen bonds (ChBs) in a set of selenium-containing molecules is a good work which was competently carried out and can be published in Molecules. I have just a few minor matters that the authors should consider before publication proceeds.
a) According to the computed values, the stabilization energies of ChBs are rather weak (less than 3 kcal/mol). We may wonder if the DFT method is the right tool to compute such small energies. It might indeed, having included long range effects in the computations. Still, it would have been good to perform a couple of post-Hartree-Fock calculations on selected examples to be sure.
b) Minor points and typos.
P.2, l. 63 and p. 3, l. 84. The chemical formulae of LAWBIV [19] and MIHFAK [20] should be given. The same for HEQTAZ [22], HOPFAU [23], ROCJOJ, etc.
P.2, l. 79. the Se·O distance of 3.454 A
P.4, l. 113. Finally, it is interesting that the X-ray structure of bis(6-(phenylselenyl)-6'-(phenylseleninyl-O)-(2,2'-bi-pyridine))-ruthenium(II) (YUTTOX) shows chloride counterions located opposite to the coordinated Se=O bonds (Figure 4c).
P.5, l. 132. Scheme 1 in bold.
P. 6, l. 149. becomes less negative (–22.6 kcal/mol),
P. 6, l. 170. DFT energetic and geometric study
P. 10, Conclusions. The (weak) value order of ChBs could be reminded in the conclusion.
Author Response
First of all, we would like to thank this reviewer for his/her careful reading of the manuscript, corrections and suggestions. The changes made and our replies follow:
This paper which theoretically studies the chalcogen bonds (ChBs) in a set of selenium-containing molecules is a good work which was competently carried out and can be published in Molecules. I have just a few minor matters that the authors should consider before publication proceeds.
- a) According to the computed values, the stabilization energies of ChBs are rather weak (less than 3 kcal/mol). We may wonder if the DFT method is the right tool to compute such small energies. It might indeed, having included long range effects in the computations. Still, it would have been good to perform a couple of post-Hartree-Fock calculations on selected examples to be sure.
Reply: Thank you for this suggestion. We have computed post HF energies (MP2) for one of the Lewis bases, giving comparable values of energies and distances. The new values have been included in Table 1 and discussed in the main text (see lines
- b) Minor points and typos.
P.2, l. 63 and p. 3, l. 84. The chemical formulae of LAWBIV [19] and MIHFAK [20] should be given. The same for HEQTAZ [22], HOPFAU [23], ROCJOJ, etc.
Reply: Done, thanks! See new Scheme 1.
P.2, l. 79. the Se·O distance of 3.454 A
Reply: Done, thanks!
P.4, l. 113. Finally, it is interesting that the X-ray structure of bis(6-(phenylselenyl)-6'-(phenylseleninyl-O)-(2,2'-bi-pyridine))-ruthenium(II) (YUTTOX) shows chloride counterions located opposite to the coordinated Se=O bonds (Figure 4c).
Reply: Fixed, thanks
P.5, l. 132. Scheme 1 in bold.
Reply: Fixed, thanks
- 6, l. 149. becomes less negative (–22.6 kcal/mol),
Reply: Fixed, thanks
- 6, l. 170. DFT energetic and geometric study
Reply: Fixed, thanks
- 10, Conclusions. The (weak) value order of ChBs could be reminded in the conclusion.
Reply: Done, thanks!